# DETR3D: 3D Object Detection
# from Multi-view Images via 3D-to-2D Queries

**Yue Wang**
Massachusetts Institute of Technology
yuewang@csail.mit.edu

**Vitor Guizilini***
Toyota Research Institute
vitor.guizilini@tri.global

**Tianyuan Zhang***
Carnegie Mellon University
tianyuaz@andrew.cmu.edu

**Yilun Wang**
Li Auto
yilunw@cs.stanford.edu

**Hang Zhao ¶**
Tsinghua University
hangzhao@mail.tsinghua.edu.cn

**Justin Solomon ¶**
Massachusetts Institute of Technology
jsolomon@mit.edu

**Abstract:** We introduce a framework for multi-camera 3D object detection. In contrast to existing works, which estimate 3D bounding boxes directly from monocular images or use depth prediction networks to generate input for 3D object detection from 2D information, our method manipulates predictions directly in 3D space. Our architecture extracts 2D features from multiple camera images and then uses a sparse set of 3D object queries to index into these 2D features, linking 3D positions to multi-view images using camera transformation matrices. Finally, our model makes a bounding box prediction per object query, using a set-to-set loss to measure the discrepancy between the ground-truth and the prediction. This top-down approach outperforms its bottom-up counterpart in which object bounding box prediction follows per-pixel depth estimation, since it does not suffer from the compounding error introduced by a depth prediction model. Moreover, our method does not require post-processing such as non-maximum suppression, dramatically improving inference speed. We achieve state-of-the-art performance on the nuScenes autonomous driving benchmark.

## 1 Introduction

3D object detection from visual information is a long-standing challenge for low-cost autonomous driving systems. While object detection from point clouds collected using modalities like LiDAR benefits from information about the 3D structure of visible objects, the camera-based setting is even more ill-posed, since we must generate 3D bounding box predictions solely from the 2D information contained in RGB images.

Existing methods [1, 2] typically build their detection pipelines purely from 2D computations. That is, they predict 3D information like object pose and velocity using an object detection pipeline designed for 2D tasks (e.g., CenterNet [1], FCOS [3]), without considering 3D scene structure or sensor configuration. These methods require several post-processing steps to fuse predictions across cameras and to remove redundant boxes, yielding a steep trade-off between efficiency and effectiveness. As an alternative to these 2D-based methods, some methods incorporate more 3D computations into our object detection pipeline by applying a 3D reconstruction method like [4, 5, 6] to create a pseudo-LiDAR or range input of the scene from camera images. Then, they could apply 3D object detection methods to this data as if it were collected directly from a 3D sensor. This strategy, however, is subject to compounding errors [7]: poorly-estimated depth values have a strongly negative effect on the performance of 3D object detection, which also can exhibit errors of its own.

---

*: Equal contribution. ¶: Co-advise on the project.

5th Conference on Robot Learning (CoRL 2021), London, UK.

In this paper, we propose a more graceful transition between 2D observations and 3D predictions for autonomous driving, which does not rely on a module for dense depth prediction. Our framework, termed DETR3D (Multi-View 3D Detection), addresses this problem in a top-down fashion. We link 2D feature extraction and 3D object prediction via geometric back-projection with camera transformation matrices. Our method starts from a sparse set of object priors, shared across the dataset and learned end-to-end. To gather scene-specific information, we back-project a set of reference points decoded from these object priors to each camera and fetch the corresponding image features extracted by a ResNet backbone [8]. The features collected from the image features of the reference points then interact with each other through a multi-head self-attention layer [9]. After a series of self-attention layers, we read off bounding box parameters from every layer and use a set-to-set loss inspired by DETR [10] to evaluate performance.

Our architecture does not perform point cloud reconstruction or explicit depth prediction from images, making it robust to errors in depth estimation. Moreover, our method does not require any post-processing, such as non-maximum suppression (NMS), improving efficiency and reducing reliance on hand-designed methods for cleaning its output. On the nuScenes dataset, our method (without NMS) is comparable with prior art (with NMS). In the camera overlap regions, our method significantly outperforms others.

**Contributions.** We summarize our key contributions as follows:

- We present a streamlined 3D object detection model from RGB images. Different from existing works that combine object predictions from the different camera views in a final stage, our method fuses information from all the camera views in each layer of computation. To the best of our knowledge, this is the first attempt to cast multi-camera detection as 3D set-to-set prediction.
- We introduce a module that connects 2D feature extraction and 3D bounding box prediction via backward geometric projection. It does not suffer from inaccurate depth predictions from a secondary network, and seamlessly uses information from multiple cameras by back-projecting 3D information onto all available frames.
- Similarly to Object DGCNN [11], our method does not require post-processing such as per-image or global NMS, and it is on par with existing NMS-based methods. In the camera overlap regions, our method outperforms others by a substantial margin.
- We release our code to facilitate reproducibility and future research.

## 2 Related Work

**2D object detection.** RCNN [12] pioneered object detection using deep learning. It feeds a set of pre-selected object proposals into a convolutional neural network (CNN) and predicts bounding box parameters accordingly. Although this method exhibits surprising performance, it is an order of magnitude slower than others because it performs a ConvNet forward pass for each object proposal. To fix this issue, Fast RCNN [13] introduces a shared learnable CNN to process the entire image at a single forward pass. To further improve performance and speed, Faster RCNN [13] includes a region proposal network (RPN) that shares full-image convolutional features with the detection network, thus enabling nearly cost-free region proposals. Mask RCNN [14] incorporates a mask prediction branch to enable instance segmentation in parallel. These methods typically involve multi-stage refinements and can be slow in practice. Different from these multi-stage methods, SSD [15] and YOLO [16] perform dense predictions in a single shot. Although they are significantly faster than the alternatives above, they still rely on NMS to remove redundant box predictions. These methods predict bounding boxes w.r.t. pre-defined anchors. CenterNet [1] and FCOS [3] change the paradigm by shifting from per-anchor prediction to per-pixel prediction, significantly simplifying the common object detection pipeline.

**Set-based object detection.** DETR [10] casts object detection as a set-to-set problem. It employs a Transformer [9] to capture feature and object interactions. DETR learns to assign predictions to a set of ground-truth boxes; thus, it does not require post-processing to filter out redundant boxes. One critical drawback of DETR, however, is that it requires a significant amount of training time. Deformable DETR [17] analyzes DETR's slow convergence and proposes a deformable self-attention module to localize features and accelerate training. Concurrently, [18] attributes the slow convergence of DETR to the set-based loss and the Transformer cross attention mechanism. They pro-

---

https://github.com/WangYueFt/detr3d

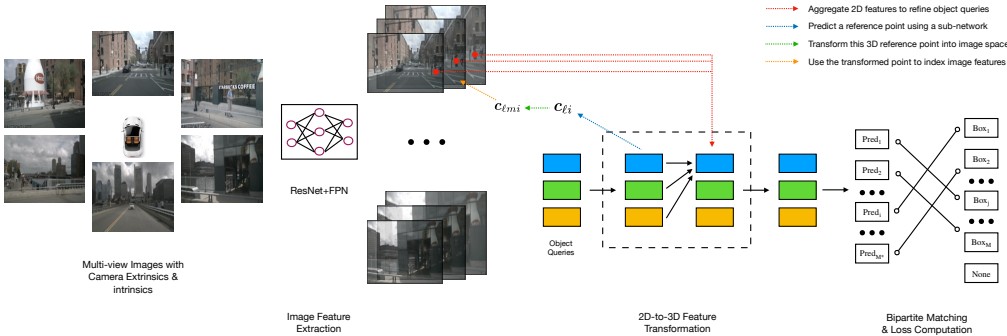

Figure 1: Overview of our method. The inputs to the model are a set of multi-view images, which are encoded by a ResNet and a FPN. Then, our model operates on a set of sparse object queries in which each query is decoded to a 3D reference point. 2D features are transformed to refine the object queries by projecting the 3D reference point into the image space. Our model makes per-query predictions and uses a set-to-set loss.

pose two variants, TSP-FCOS and TSP-RCNN, to overcome these problematic aspects. SparseR-CNN [19] incorporates set prediction into a RCNN-style pipeline; it outperforms multi-stage object detection without NMS. OneNet [20] studies an interesting phenomenon: dense-based object detectors can be made NMS-free after they are equipped with a minimum-cost set loss. For 3D domains, Object DGCNN [11] studies 3D object detection from point clouds. It models 3D object detection as message passing on a dynamic graph, generalizing the DGCNN framework to predict a set of objects. Similar to DETR, Object DGCNN is also NMS-free.

**Monocular 3D object detection.** An early method for 3D detection from RGB images is Mono3D [21], which uses semantic and shape cues to select from a collection of 3D proposals, using scene constraints and additional priors at training time. [22] uses the birds-eye-view (BEV) for monocular 3D detection, and [23] leverages 2D detections for 3D bounding box regression via the minimization of 2D-3D projection error. The use of 2D detectors as a starting point for 3D computation recently has become a standard approach [24, 25]. Other works also explore advances in differentiable rendering [26] or 3D keypoint detection [27, 28, 1] to enable state-of-the-art 3D object detection performance. All these methods operate in a monocular setting, and extensions to multiple cameras are done by independently processing each frame before merging the outputs in a post-processing stage.

## 3  Multi-view 3D Object Detection

### 3.1  Overview

Our architecture inputs RGB images collected from a set of cameras whose projection matrices (the combination of intrinsics and relative extrinsics) are known, and it outputs a set of 3D bounding box parameters for the objects in the scene. In contrast to past approaches, we build our architecture based on a few high-level desiderata:

- We incorporate 3D information into intermediate computations within our architecture, rather than performing purely 2D computations in the image plane.
- We do not estimate dense 3D scene geometry, avoiding associated reconstruction errors.
- We avoid post-processing steps such as NMS.

We address these desiderata using a new set prediction module, which links 2D feature extraction and 3D box prediction by alternating between 2D and 3D computations. Our model contains three critical components, illustrated in Figure 1. First, following common practice in 2D vision, it extracts features from the camera images using a shared ResNet [8] backbone. Optionally, these features are enhanced by a feature pyramid network (FPN) [29] (§3.2). Second, a detection head (§3.3)—our main contribution—links the computed 2D features to a set of 3D bounding box predictions in a geometry-aware manner (§3.3). Each layer of the detection head starts from a sparse set of object queries, which are learned from the data. Each object query encodes a 3D location, which

is projected to the camera planes and used to collect image features via bilinear interpolation. Similarly to DETR [10], we then use multi-head attention [9] to refine the object queries by incorporating object interactions. This layer is repeated multiple times, alternating between feature sampling and object query refinement. Finally, we evaluate a set-to-set loss [30, 10] to train the network (§3.4).

## 3.2 Feature Learning

Our model starts with a set of images $\mathcal{I} = \{\mathbf{im}_1, \dots, \mathbf{im}_K\} \subset \mathbb{R}^{H_{\mathrm{im}} \times W_{\mathrm{im}} \times 3}$ (captured by surrounding cameras), camera matrices $\mathcal{T} = \{T_1, \dots, T_K\} \subset \mathbb{R}^{3 \times 4}$, ground-truth bounding boxes $\mathcal{B} = \{\boldsymbol{b}_1, \dots, \boldsymbol{b}_j, \dots, \boldsymbol{b}_M\} \subset \mathbb{R}^9$, and categorical labels $\mathcal{C} = \{c_1, \dots, c_j, \dots, c_M\} \subset \mathbb{Z}$. Each $\boldsymbol{b}_j$ contains position, size, heading angle, and velocity in the birds-eye view (BEV); our model aims to predict these boxes and their labels from the these images. We *do not* use point clouds, which are usually captured by high-end LiDAR.

These images are encoded with a ResNet [8] and a FPN [29] into four sets of features $\mathcal{F}_1, \mathcal{F}_2, \mathcal{F}_3, \mathcal{F}_4$. Each set $\mathcal{F}_k = \{\boldsymbol{f}_{k1}, \dots, \boldsymbol{f}_{k6}\} \subset \mathbb{R}^{H \times W \times C}$ corresponds to a level of features of the 6 images. These multi-scale features provide rich information to recognize objects of different sizes. Next, we detail our approach to transform these 2D features into 3D using a novel set prediction module.

## 3.3 Detection Head

Existing methods for detecting objects from camera input typically employ a bottom-up approach, which predicts a dense set of bounding boxes per image, filters redundant boxes between the images, and aggregates predictions across cameras in a post-processing step. This paradigm has two crucial drawbacks: dense bounding box prediction requires accurate depth perception, which itself is a challenging problem; and NMS-based redundancy removal and aggregation are non-parallelizable operations that introduce significant inference overhead. We address these issues using a top-down object detection head described below.

Analogously to [11, 17], DETR3D is *iterative*; it uses $L$ layers with set-based computations to produce bounding box estimates from 2D feature maps. Each layer includes the following steps:

1. predict a set of bounding box centers associated with object queries;
2. project these centers into all the feature maps using the camera transformation matrices;
3. sample features via bilinear interpolation and incorporate them into object queries; and
4. describe object interactions using multi-head attention.

Motivated by DETR [10], each layer $\ell \in \{0, \dots, L - 1\}$ operates on a set of *object queries* $\mathcal{Q}_\ell = \{\boldsymbol{q}_{\ell 1}, \dots, \boldsymbol{q}_{\ell M^*}\} \subset \mathbb{R}^C$, producing a new set $\mathcal{Q}_{\ell+1}$. A reference point $\boldsymbol{c}_{\ell i} \in \mathbb{R}^3$ is decoded from a object query $\boldsymbol{q}_{\ell i}$ as follows:

$$\boldsymbol{c}_{\ell i} = \Phi^{\mathrm{ref}}(\boldsymbol{q}_{\ell i}), \tag{1}$$

where $\Phi^{\mathrm{ref}}$ is a neural network. $\boldsymbol{c}_{\ell i}$ can be thought of a hypothesis for the center of the $i$-th box. Next, we acquire image features corresponding to $\boldsymbol{c}_{\ell i}$ to refine and predict the final bounding box. Then, $\boldsymbol{c}_{\ell i}$ (or more accurately its homogeneous counterpart $\boldsymbol{c}_{\ell i}^*$) is projected into each one of the images using the camera transformation matrices:

$$\boldsymbol{c}_{\ell i}^* = \boldsymbol{c}_{\ell i} \oplus 1 \qquad \boldsymbol{c}_{\ell m i} = T_m \boldsymbol{c}_{\ell i}^*, \tag{2}$$

where $\oplus$ denotes concatenation, and $\boldsymbol{c}_{\ell m i}$ is the projection of the reference point onto the $m$-th camera. To remove the effects of the feature map size and gather features across different levels, we normalize $\boldsymbol{c}_{\ell m i}$ to $[-1, 1]$. Next, the images features are collected by

$$\boldsymbol{f}_{\ell k m i} = f^{\mathrm{bilinear}}(\mathcal{F}_{km}, \boldsymbol{c}_{\ell m i}), \tag{3}$$

where $\boldsymbol{f}_{\ell k m i}$ is the feature for $i$-th point from $k$-th level of $m$-th camera at $\ell$-th layer.

A given reference point is not necessarily visible in all the camera images, so we need some heuristics to filter invalid points. To that end, we define a binary value $\sigma_{\ell k m i}$, which is determined based on whether a reference point is projected outside an image plane. The final feature $\boldsymbol{f}_{\ell i}$ and object query in next layer $\boldsymbol{q}_{(\ell+1)i}$ are given by

$$\boldsymbol{f}_{\ell i} = \frac{1}{\sum_k \sum_m \sigma_{\ell k m i} + \epsilon} \sum_k \sum_m \boldsymbol{f}_{\ell k m i} \sigma_{\ell k m i} \qquad \text{and} \qquad \boldsymbol{q}_{(\ell+1)i} = \boldsymbol{f}_{\ell i} + \boldsymbol{q}_{\ell i}, \tag{4}$$

where $\epsilon$ is a small number to avoid division by zero. Finally, for each object query $\boldsymbol{q}_{\ell i}$, we predict a bounding box $\hat{\boldsymbol{b}}_{\ell i}$ and its categorical label $\hat{c}_{\ell i}$ with two neural networks $\Phi_\ell^{\mathrm{reg}}$ and $\Phi_\ell^{\mathrm{cls}}$:

$$\hat{\boldsymbol{b}}_{\ell i} = \Phi_\ell^{\mathrm{reg}}(\boldsymbol{q}_{\ell i}) \qquad \text{and} \qquad \hat{c}_{\ell i} = \Phi_\ell^{\mathrm{cls}}(\boldsymbol{q}_{\ell i}). \tag{5}$$

We compute the loss for the predictions $\hat{\mathcal{B}}_\ell = \{\hat{\boldsymbol{b}}_{\ell 1}, \ldots, \hat{\boldsymbol{b}}_{\ell j}, \ldots, \hat{\boldsymbol{b}}_{\ell M^*}\} \subset \mathbb{R}^9$ and $\hat{\mathcal{C}}_\ell = \{\hat{c}_{\ell 1}, \ldots, \hat{c}_{\ell j}, \ldots, \hat{c}_{\ell M}\} \subset \mathbb{Z}$ from every layer during training. During inference, we only use the outputs from the last layer.

## 3.4  Loss

Following [30, 10], we use a set-to-set loss to measure the discrepancy between the prediction set $(\hat{\mathcal{B}}_\ell, \hat{\mathcal{C}}_\ell)$ and the ground-truth set $(\mathcal{B}, \mathcal{C})$. This loss consists of two parts: a focal loss [31] for the class labels and a $L^1$ loss for the bounding box parameters. For notational convenience, we drop the $\ell$ subscript in $\hat{\mathcal{B}}_\ell$ and $\hat{\mathcal{C}}_\ell$. The number of ground-truth boxes $M$ is typically smaller than the number of predictions $M^*$, so we pad the set of ground-truth boxes with $\varnothing$s (no object) up to $M^*$ for ease of computation. We establish a correspondence between the ground-truth and the prediction via a bi-partite matching problem: $\sigma^* = \arg\min_{\sigma \in \mathcal{P}} \sum_{j=1}^M -\mathbb{1}_{\{c_j \neq \varnothing\}} \hat{p}_{\sigma(j)}(c_j) + \mathbb{1}_{\{c_j \neq \varnothing\}} \mathcal{L}_{\mathrm{box}}(\boldsymbol{b}_j, \hat{\boldsymbol{b}}_{\sigma(j)})$,

where $\mathcal{P}$ denotes the set of permutations, $\hat{p}_{\sigma(j)}(c_j)$ is the probability of class $c_j$ for the prediction with index $\sigma(j)$, and $\mathcal{L}_{\mathrm{box}}$ is the $L_1$ loss for bounding box parameters. We use the Hungarian algorithm [32] to solve this assignment problem, as in [30, 10], yielding the set-to-set loss $\mathcal{L}_{\mathrm{sup}} = \sum_{j=1}^N -\log \hat{p}_{\sigma^*(j)}(c_j) + \mathbb{1}_{\{c_j \neq \varnothing\}} \mathcal{L}_{\mathrm{box}}(\boldsymbol{b}_j, \hat{\boldsymbol{b}}_{\sigma^*(j)})$.

## 4  Experiments

We present our results as follows: first, we detail the dataset, metrics, and implementation in §4.1; then we compare our method to existing works in §4.2; we benchmark the performance of different models in camera overlap regions in §4.3; we compare to a forward prediction model in §4.4; and we provide additional analysis and ablations in §4.5.

### 4.1  Implementation Details

**Dataset.** We test our method on the nuScenes dataset [33]. nuScenes consists of 1,000 sequences; each sequence is roughly 20s long, with a sampling rate of 20 frames/second. Each sample contains images from 6 cameras [`front_left`, `front`, `front_right`, `back_left`, `back`, `back_right`]. Camera parameters including intrinsics and extrinsics are available. nuScenes provides annotations every 0.5s; in total there are 28k, 6k, and 6k annotated samples for training, validation, and testing, respectively. 10 from the total 23 classes are available to compute the metrics.

**Metrics.** We follow the official evaluation protocol provided by nuScenes. We evaluate average translation error (ATE), average scale error (ASE), average orientation error (AOE), average velocity error (AVE), and average attribute error (AAE). These metrics are true positive metrics (TP metrics) and computed in the physical unit. In addition, we measure mean average precision (mAP). To capture all aspects of the detection task, a consolidated scalar metric–the nuScenes Detection Score (NDS) [33]–is defined as $\mathrm{NDS} = \frac{1}{10}[5\mathrm{mAP} + \sum_{\mathrm{mTP} \in \mathbb{TP}}(1 - \min(1, \mathrm{mTP}))]$.

**Model.** Our model consists of a ResNet [8] feature extractor, a FPN, and a DETR3D detection head. We use ResNet101 with deformable convolutions [34] in the 3rd stage and 4th stage. The FPN [29] takes features output by the ResNet and produces 4 feature maps whose sizes are $1/8$, $1/16$, $1/32$, and $1/64$ of the input image sizes. The DETR3D detection head consists of 6 layers, where each layer is a combination of a feature refinement step and a multi-head attention layer. The hidden dimension of the DETR3D detection head is 256. Finally, two sub-networks predict bounding box parameters and a class label per object query; each sub-network consists of two fully connected layers with hidden dimensions 256. We use LayerNorm [35] in the detection head.

**Training & inference.** We use AdamW [36] to train the whole pipeline. The weight decay is $10^{-4}$. We use an initial learning rate $10^{-4}$, which is decreased to $10^{-5}$ and $10^{-6}$ at 8th and 11th epochs. The model is trained for 24 epochs in total on 8 RTX 3090 GPUs and the per-GPU batch size is 1.

Table 1: Comparisons to recent works on the validation set. Our method is robust to the usage of NMS. ∗: CenterNet uses a customized backbone DLA [38]. ‡: this model is trained with depth weight 1.0 and initialized from a FCOS3D checkpoint; the checkpoint is trained on the same dataset with depth weight 0.2. §: with test-time augmentation. ¶: with test-time augmentation, more epochs, and model ensemble. For details, see [2]. †: our model is also initialized from a FCOS3D backbone; the detection head is initialized randomly. #: trained with CBGS [39]

| Method | NDS ↑ | mAP ↑ | mATE ↓ | mASE ↓ | mAOE ↓ | mAVE ↓ | mAAE ↓ | NMS |
|--------|-------|-------|--------|--------|--------|--------|--------|-----|
| CenterNet ∗ | 0.328 | 0.306 | 0.716 | 0.264 | 0.609 | 1.426 | 0.658 | ✓ |
| FCOS3D | 0.373 | 0.299 | 0.785 | 0.268 | 0.557 | 1.396 | 0.154 | ✓ |
| FCOS3D ‡ | 0.393 | 0.321 | 0.746 | 0.265 | 0.503 | 1.351 | 0.160 | ✓ |
| FCOS3D § | 0.402 | 0.326 | 0.743 | 0.259 | 0.441 | 1.341 | 0.163 | ✓ |
| FCOS3D ¶ | 0.415 | 0.343 | 0.725 | 0.263 | 0.422 | 1.292 | 0.153 | ✓ |
| DETR3D (Ours) | 0.374 | 0.303 | 0.860 | 0.278 | 0.437 | 0.967 | 0.235 | - |
| DETR3D (Ours) † | 0.425 | 0.346 | 0.773 | 0.268 | 0.383 | 0.842 | 0.216 | - |
| DETR3D (Ours) # | 0.434 | 0.349 | 0.716 | 0.268 | 0.379 | 0.842 | 0.200 | - |

Table 2: Comparisons to top-performing works on the test set from the leaderboard. #: initialized from a DD3D checkpoint. †: initialized from a backbone pre-trained on extra data.

| Method | NDS ↑ | mAP ↑ | mATE ↓ | mASE ↓ | mAOE ↓ | mAVE ↓ | mAAE ↓ | NMS |
|--------|-------|-------|--------|--------|--------|--------|--------|-----|
| Mono3D | 0.429 | 0.366 | 0.642 | 0.252 | 0.523 | 1.591 | 0.119 | N/A |
| DHNet | 0.437 | 0.363 | 0.667 | 0.259 | 0.402 | 1.589 | 0.120 | N/A |
| PGD [40] | 0.448 | 0.386 | 0.626 | 0.245 | 0.451 | 1.509 | 0.127 | ✓ |
| DD3D [37] † | 0.477 | 0.418 | 0.572 | 0.249 | 0.368 | 1.014 | 0.124 | ✓ |
| DETR3D (Ours) # | 0.479 | 0.412 | 0.641 | 0.255 | 0.394 | 0.845 | 0.133 | - |

The training procedure takes roughly 18 hours. We do not use any post-processing such as NMS during inference. For evaluation, we use the nuScenes evalutation toolkit.

## 4.2 Comparison to Existing Works

We compare to previous state-of-the-art methods CenterNet [1] and FCOS3D [2]. CenterNet is an anchor-free 2D detection method that makes dense predictions in a high resolution feature map. FCOS3D employs a FCOS [3] pipeline to make per-pixel predictions. These methods both turn 3D object detection into a 2D problem, and in doing so ignore scene geometry and sensor configuration. To perform multi-view object detection, these methods have to process each image independently, and use both per-image and global NMS to remove redundant boxes in each view and in the overlap regions respectively. As shown in Table 1, our method outperforms these methods even though we do not use any post-processing. Our method performs worse than FCOS3D in terms of mATE. We suspect this is because FCOS3D directly predicts bounding box depth, which leads to strong supervision on object translation. Also, FCOS3D uses disentangled heads for different bounding box parameters, which can increase performance.

On the test set (Table 2), our method outperforms all existing methods as of 10/13/2021; our method uses the same backbone as DD3D [37] for a fair comparison.

## 4.3 Comparison in Overlap Regions

A great challenge lies in the overlap regions where objects are more likely to be cut off. Our method considers all cameras simultaneously, while FCOS3D predicts bounding boxes per camera individually. To further demonstrate the advantages of fused inference, we calculate the metrics for boxes falling into the camera overlaps. To compute the metrics, we select boxes whose 3D center is visible to multiple cameras. On the validation set, there are 18,147 such boxes, 9.7% of the total. Table 3 shows the results; our method outperforms FCOS3D remarkably in terms of NDS scores in this setting. This confirms that our integrated prediction approach is more effective.

Table 3: Comparisons in Overlap Region. ‡: this model is trained with depth weight 1.0 and initialized from a FCOS3D checkpoint; the checkpoint is trained on the same dataset with depth weight 0.2. For details, see [2]. †: our model is also initialized from a FCOS3D backbone; the detection head is initialized randomly.

| Method | NDS ↑ | mAP ↑ | mATE ↓ | mASE ↓ | mAOE ↓ | mAVE ↓ | mAAE ↓ | NMS |
|--------|-------|-------|--------|--------|--------|--------|--------|-----|
| FCOS3D | 0.317 | 0.213 | 0.841 | 0.276 | 0.604 | 1.122 | 0.173 | ✓ |
| FCOS3D‡ | 0.329 | 0.229 | 0.816 | 0.272 | 0.571 | 1.084 | 0.195 | ✓ |
| DETR3D (Ours) | 0.356 | 0.231 | 0.825 | 0.280 | 0.400 | 0.863 | 0.223 | - |
| DETR3D (Ours) † | 0.384 | 0.268 | 0.807 | 0.273 | 0.453 | 0.788 | 0.184 | - |

Table 4: Comparisons to pseudo-LiDAR Methods.

| Method | NDS ↑ | mAP ↑ | mATE ↓ | mASE ↓ | mAOE ↓ | mAVE ↓ | mAAE ↓ | NMS |
|--------|-------|-------|--------|--------|--------|--------|--------|-----|
| pseudo-LiDAR | 0.160 | 0.048 | - | - | - | - | - | ✓ |
| DETR3D (Ours) | 0.374 | 0.303 | 0.860 | 0.278 | 0.437 | 0.967 | 0.235 | - |

### 4.4 Comparison to pseudo-LiDAR Methods

Another way to perform 3D object detection is by generating pseudo-LiDAR point clouds from multi-view images using a depth prediction model. On the nuScenes dataset, there are no publicly available pseudo-LiDAR works for us to make a direct comparison. Hence, we implement a baseline ourselves to verify that our approach is more effective than explicit depth prediction. We use a pre-trained PackNet [6] network to predict dense depth maps from all six cameras and then convert these depth maps into point clouds using the camera transformations. We also experimented with a self-supervised PackNet model with velocity supervision (as in the original paper), but we found that ground-truth depth supervision yielded more realistic point clouds and therefore used a supervised model as baseline. For 3D detection, we employ the recently-proposed CenterPoint architecture [41]. Conceptually, this pipeline is a variant of pseudo-LiDAR [42]. Table 4 shows the results; we conclude that this pseudo-LiDAR method underperforms ours significantly even when depth estimates are generated by a state-of-the-art model. One possible explanation is that pseudo-LiDAR object detectors suffer from compounding errors introduced by inaccurate depth prediction, that in turn is known to overfit to training data and generalizes poorly to other distributions [7].

### 4.5 Ablation & Analysis

We provide a visualization of object query refinement in Figure 2. We visualize bounding boxes decoded from the object queries in each layer. The predicted bounding boxes get closer to the ground-truth as we go into deeper layers in the model.Also, the leftmost figure shows the learned object query priors shared by all data. We also provide quantitative results in Table 5, which shows that iterative refinement indeed improves performance significantly. This suggests that iterative refinement is both beneficial and necessary to fully leverage our proposed architecture. Furthermore, we provide ablations on the number of object queries in Table 6; increasing the number queries consistently improves the performance until it gets saturated at 900. Finally, Table 7 shows the results with different backbones.

## 5 Conclusion

We propose a new paradigm to address the ill-posed inverse problem of recovering 3D information from 2D images. In this setting, the input signal lacks essential information for models to make effective predictions without priors learned from data. While other methods either operate solely on 2D computations or use additional depth networks to reconstruct the scene, ours operates in 3D space and uses backward projection to retrieve image features as needed. The benefits of our approach are two-fold: (1) it eliminates the need for middle-level representations (e.g., predicted depth maps or point clouds), which can be a source of compounding errors; and (2) it uses information from multiple cameras by projecting the same 3D point onto all available frames.

Table 5: Evaluation on detection results from different layers.

| Layer ↑ | NDS ↑ | mAP ↑ | mATE ↓ | mASE ↓ | mAOE ↓ | mAVE ↓ | mAAE ↓ |
|---|---|---|---|---|---|---|---|
| 0 | 0.380 | 0.302 | 0.855 | 0.280 | 0.435 | 0.910 | 0.231 |
| 1 | 0.410 | 0.335 | 0.791 | 0.275 | 0.408 | 0.887 | 0.217 |
| 2 | 0.420 | 0.343 | 0.782 | 0.271 | 0.395 | 0.851 | 0.214 |
| 3 | 0.420 | 0.346 | 0.778 | 0.268 | 0.390 | 0.874 | 0.218 |
| 4 | 0.424 | 0.346 | 0.777 | 0.268 | 0.389 | 0.855 | 0.217 |
| 5 | 0.425 | 0.346 | 0.773 | 0.268 | 0.383 | 0.842 | 0.216 |

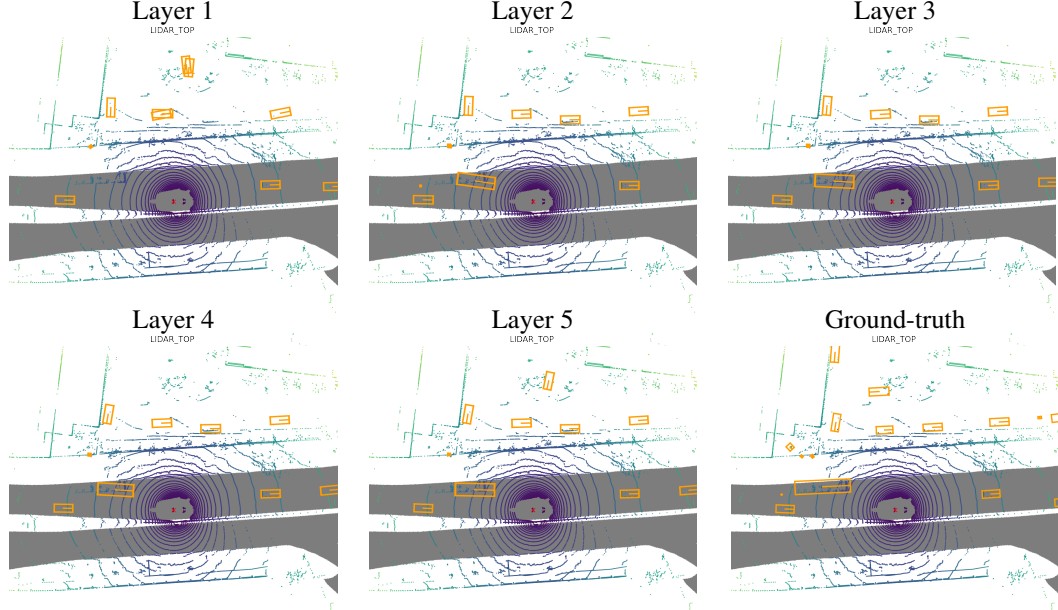

Figure 2: Detection results from layer 1 to layer 5 in the DETR3D head. We visualize the bounding boxes in the BEV and overlay the point clouds from lidar_top. The predictions get closer to the ground-truth in the deeper layers.

Table 6: Results with different number of queries.

| # queries | 30 | 100 | 300 | 600 | 900 | 1200 | 1500 |
|---|---|---|---|---|---|---|---|
| mAP ↑ | 0.201 | 0.313 | 0.338 | 0.347 | 0.346 | 0.340 | 0.346 |
| NDS ↑ | 0.331 | 0.408 | 0.415 | 0.420 | 0.425 | 0.415 | 0.420 |

Table 7: Results with different backbones.

| Backbone ↑ | NDS ↑ | mAP ↑ | mATE ↓ | mASE ↓ | mAOE ↓ | mAVE ↓ | mAAE ↓ |
|---|---|---|---|---|---|---|---|
| ResNet50 | 0.373 | 0.302 | 0.811 | 0.282 | 0.493 | 0.979 | 0.212 |
| ResNet101 | 0.425 | 0.346 | 0.773 | 0.268 | 0.383 | 0.842 | 0.216 |
| DLA34 | 0.394 | 0.312 | 0.829 | 0.276 | 0.450 | 0.844 | 0.221 |

Beyond the direct application of our work to 3D object detection for autonomous driving, there are several venues that warrant future investigation. For example, single point projection creates a limited receptive field in the retrieved image feature maps, and sampling multiple points for each object query would incorporate more information for object refinement. Furthermore, the new detection head is input-agnostic, and including other modalities such as LiDAR/RADAR would enhance performance and robustness. Finally, generalizing our pipeline to other domains such as indoor navigation and object manipulation would increase its scope of application and reveal additional ways for further improvement.

## Acknowledgement

The MIT Geometric Data Processing group acknowledges the generous support of Army Research Office grants W911NF2010168 and W911NF2110293, of Air Force Office of Scientific Research award FA9550-19-1-031, of National Science Foundation grants IIS-1838071 and CHS-1955697, from the CSAIL Systems that Learn program, from the MIT–IBM Watson AI Laboratory, from the Toyota–CSAIL Joint Research Center, from a gift from Adobe Systems, from an MIT.nano Immersion Lab/NCSOFT Gaming Program seed grant, and from the Skoltech–MIT Next Generation Program.

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
