# OpenReview forum: "DETR3D: 3D Object Detection from Multi-view Images via 3D-to-2D Queries"
_robot-learning.org/CoRL/2021/Conference — CoRL2021 Poster_

### Official Review · Reviewer_KAPs · 2021-07-22

**Originality:** Good
**Technical Quality:** Fair
**Clarity Of Presentation:** Very Good
**Impact:** 3

**Recommendation:**

Weak Accept: I recommend accepting the paper, but will not argue for my recommendation if the majority of other reviewers have a different opinion.

**Summary:**

This paper introduces a detection head for 3D bounding box object in multi-view images. The approach is based on a DETR-like architecture, where object queries are iteratively passed through transformer layers to generate predictions and trained with a set-to-set loss. The key technique proposed in this work is to predict 3D bounding box centers in the object queries that are then backprojected to the images to gather multi-view features before further refinening them with the same process iteratively. Experiments are conducted on the nuScenes dataset where 360 degree cameras are available.

**Issues:**

I'm willing to improve my rating based on the response to my comments in "Weaknesses" above.

**Reviewer Expertise:**

Good: General knowledge of the area

**Strengths And Weaknesses:**

Strenghts:

- The multi-view feature aggregation method proposed is sensible. Specifically, performing flexible multi-view aggregation with attention seems like a good direction that can suppress spurious errors.

- The approach is comparatively fast (although not real-time) because it doesn't require NMS.

- The paper is clear, self-contained, and overall well-written.

Weaknesses:

- The approach constrains itself to showing results in the setting of 360 degree camera coverage where there is a lack of appropriate baselines, specifically Pseudo-LiDAR. The authors resort to re-implementing a baseline in their setting, but the results indicate that the baseline works so bad that it isn't clear if this issue is due to a problem in the method itself or in the implementation. However, constraining to such a restricted setting is not necessary as far as far as I can tell. For example, when one would only use the two-view stereo setup, it would be possible to evaluate on KITTI where code for the original pseudo-LiDaR work is available. I would strongly encourage such an evaluation.

- The main evaluation compares to CenterNet and FCOS3D, both of which are intrinsically single-image methods. They require a ad-hoc postprocessing heuristic to bring them into the multi-view setting. As such these methods are fundamentally at a disadvantage. It is not clear to me if this comparison is truly fair. As such a proper comparison to the Pseudo-LiDAR line of work seems even more necessary.  Nonetheless, is seems that the baselines, and specifically FCOS3D, perform rather competitive when looking at Table 1. Table 2 suggests a strong advantage of the proposed method in the overlap regions, which seems reasonable given the setting, but also suggest together with Table 1 that the approach is at a strong disadvantage in non-overlap regions.

- The proposed approach foregoes explicit depth estimation and directly regresses 3D bounding box centers which are used for multi-view feature aggregations. The authors state that this is beneficial, as imperfect depth estimates can negatively influence results. However, the task of direct regression seems harder than it needs to be. Some form of depth is always available in this the multi-view setting, even if it might be imperfect. I believe the claims about compounding error in the depth maps could and should be considerably strengthened using more fine-grained ablations: a) use the estimated depths to perform warping and feature aggregation instead of using regressed bounding box centers. b) use the estimated depths as an additional input modality in the existing pipeline. c) potentially use "groundtruth" LiDAR depths in both a) and b) as an oracle in order to quantify the performance loss due to imperfect depth maps and imperfect estimates of 3D bounding boxes.

- The baselines in Table 1 are not clear to me. Is the DLA backbone in CenterNet and the overall architecture comparable in terms of capacity? What does it mean that FCOS3D is used without fine-tuning? On what dataset was FCOS3D trained?

Minor comments:

- The captions in Figure 1 are tiny and very hard to read.
- Figure 2: clarity could be improved by showing groundtruth bounding boxes directly in the layer visualization using a different color, so that the reader can easier see the actual differenc to the groundtruth. One thing to consider could also be color coding all the layer predictions and merging into a single image to better see the evolution of predictions.
- l. 257: should this be "upper left" instead of "leftmost"?


Typo:
- l.220: "ignore" -> "ignores"

**Summary Of Recommendation:**

Technically, the proposed approach is a relatively straight-forward, but sensible, application of the DETR detection paradigm to the task of 3D bounding detection. I believe the approach has potential, but the current evaluation has multiple issues that do not allow to position this work appropriately with respect to existing works.

-- Post rebuttal --

The authors revised the paper with additional experiments and ablations, which I believe greatly strenghtens the work. I thus raise my score to weak accept.

---

### Official Review · Reviewer_Gg7X · 2021-07-25

**Originality:** Very Good
**Technical Quality:** Fair
**Clarity Of Presentation:** Good
**Impact:** 3

**Recommendation:**

Weak Reject: I recommend rejecting the paper, but will not argue for my recommendation if the majority of other reviewers have a different opinion.

**Summary:**

The paper presents a multi-view (multi-image) 3D object detector based on transformers. The backbone architecture follows DETR, but queries are shared among multiple views. The resulting detector performs on-par with the baseline on nuScenes.

**Issues:**

Wee above.

**Reviewer Expertise:**

Very good: Comprehensive knowledge of the area

**Strengths And Weaknesses:**

+ Interesting architecture, clever use of DETR (and its object queries)
+ Easy to read and follow
- Limited ablations
- Performance not SOTA (despite claim)

The paper is well written and easy to follow. The main idea of using object queries shared between multiple images and a transformer that can exchange information between those is clever and novel.

Unfortunately, the main technical novelty is not really evaluated or ablated. The 360 degree cameras on a car offer only limited parallax, and an even smaller overlap in field of view to take advantage of. It is thus unclear if a 3D lifting is even needed, or if a simple shift and shear operations given the calibrated cameras would suffice?

Finally, the authors claim to be SOTA, but do not outperform FCOS3D. In fact, the presented method only matches the performance of the baseline FCOS3D (without fine-tuning). It seems odd to take prior work, and simply not compare to their best model.

**Summary Of Recommendation:**

The basic idea is cute and interesting, the evaluation and comparison fall short.

---

### Official Review · Reviewer_tKqW · 2021-07-25

**Originality:** Good
**Technical Quality:** Good
**Clarity Of Presentation:** Fair
**Impact:** 3

**Recommendation:**

Weak Accept: I recommend accepting the paper, but will not argue for my recommendation if the majority of other reviewers have a different opinion.

**Summary:**

The paper proposes a novel approach for multi-camera 3D object detection. The proposed model is based on DETR, a transformer-based object detection network, by extending it to multi-camera 3D detection. The main contributions compared to DETR are:
- projecting the learned object queries into each camera using the calibration matrices,
- sampling features from the projected queries and feed them to the transformed decoder,
- regressing bounding boxes in 3D instead of 2D space.

The method is evaluated on the nuScenes dataset and compared to two existing approaches.

**Issues:**

- The paper should better discuss the reason why they compared with specific versions of existing approaches, and why the results of some of them differ from the ones available on the leaderboard.
- The paper should better explain the network architecture, without relying on the readers' knowledge of DETR.
- The paper should include a runtime analysis to support the claim that the proposed approach improves efficiency.

**Reviewer Expertise:**

Good: General knowledge of the area

**Strengths And Weaknesses:**

Strengths:
+ The paper is well written and well organized.
+ The proposed approach is very interesting, especially the idea of learning object queries in the 3D space and projecting them into each camera to sample the associated features.
+ The proposed model does not require post-processing techniques, such as NMS, which is usually required by existing object detection methods.
+ Differently from existing methods, which perform multi-camera inference by processing each camera independently, the proposed approach directly fuses the features from different cameras.

Weaknesses:
- My main concern regards the experimental evaluation. In Table 1 the authors state that they "compare to the FCOS3D without fine-tuning and test-time augmentation". Since the proposed method does not use test-time augmentation (or at least is not mentioned in the paper), I can understand the comparison against FCOS3D without it. However, I do not understand why they compared against FCOS3D without fine-tuning. Moreover, the results reported for CenterNet are different from the ones reported in the nuScenes leaderboard. Both FCOS3D with fine-tuning and CenterNet leaderboard results are better than the results achieved by the proposed approach.
- The ablation study is a bit shallow, only the number of decoder layers is analyzed. More ablation studies would be appreciated, such as on the number of object queries and different feature extractors.
- For readers who are not familiar with the DETR model, the network architecture is not very clear. For example, the set of learned object queries is not explained properly, what do they represent? In general, the understanding of this paper requires a good understanding of the DETR architecture. Moreover, Fig. 1 is not very helpful in understanding the architecture.
- The paper claims that the proposed approach "does not require any post-processing, such as non-maximum suppression (NMS), improving efficiency" (L.46). Therefore, I would expect an analysis of the inference time to be reported.

Typo:
(L.132): "from the these images"

**Summary Of Recommendation:**

Weak Accept: the proposed approach is interesting, but the paper should provide a better description of the architecture, and the results are not entirely convincing.

Post rebuttal: the authors' addressed all my comments and added several comparisons and experiments that strengthen the claim of the paper. I suggest that the paper should be accepted.

---

### Meta-Review · Area_Chair_1gR3 · 2021-08-12

**Recommendation:** Accept (Poster)
**Confidence:** 4

**Metareview:**

There is a clear agreement between the reviewers that the main idea of the paper, namely the extension of the transformer-based object detector DETR to the multi-view setting, is reasonable and nice. However, there are also a number of important shortcomings. These mainly include a somewhat unbalanced comparison to the state of the art (e.g. skipping fine-tuning in FCOS3D or the two-view pseudo-LIDAR), a number of missing ablation studies (e.g. re number of queries, or estimated depth), and the not obvious motivation to use 3D lifting over simpler 2D techniques. Out of these, the first one weighs most, given that the claim of obtaining SOTA results with the proposed approach becomes questionable. A detailed discussion on this would be very helpful, maybe also including some further reasoning on the choice of experimental comparisons, or else on some potential additional benefits of the proposed approach over the existing techniques.

Post-rebuttal:
The authors made a significant effort to show that their results are an imprvement over existing apporaches, namely FCOS3D. The results seem to be of interest for the community. The authors are adviced to thoroughly describe and analyse these additional results in the final version of the paper.

---

### Decision · Program_Chairs · 2021-09-13

**Decision:**

Accept (Poster)

**Comment:**

There is a clear agreement between the reviewers that the main idea of the paper, namely the extension of the transformer-based object detector DETR to the multi-view setting, is reasonable and nice. However, there are also a number of important shortcomings. These mainly include a somewhat unbalanced comparison to the state of the art (e.g. skipping fine-tuning in FCOS3D or the two-view pseudo-LIDAR), a number of missing ablation studies (e.g. re number of queries, or estimated depth), and the not obvious motivation to use 3D lifting over simpler 2D techniques. Out of these, the first one weighs most, given that the claim of obtaining SOTA results with the proposed approach becomes questionable. A detailed discussion on this would be very helpful, maybe also including some further reasoning on the choice of experimental comparisons, or else on some potential additional benefits of the proposed approach over the existing techniques.

Post-rebuttal:
The authors made a significant effort to show that their results are an imprvement over existing apporaches, namely FCOS3D. The results seem to be of interest for the community. The authors are adviced to thoroughly describe and analyse these additional results in the final version of the paper.